# Yoga Meets Intelligent Internet of Things: Recent Challenges and Future Directions

**DOI:** 10.3390/bioengineering10040459

**Published:** 2023-04-09

**Authors:** Rishi Pal, Deepak Adhikari, Md Belal Bin Heyat, Inam Ullah, Zili You

**Affiliations:** 1Center of Psychosomatic Medicine, Sichuan Provincial Center for Mental Health, School of Life Science and Technology, University of Electronic Science and Technology of China, Chengdu 610054, China; 2School of Information and Software Engineering, University of Electronic Science and Technology of China, Chengdu 610056, China; 3IoT Research Center, College of Computer Science and Software Engineering, Shenzhen University, Shenzhen 518060, China; 4Department of Computer Engineering, Gachon University, Sujeong-gu, Seongnam 13120, Republic of Korea

**Keywords:** future intelligence, health intelligence, IoT, IoMT, medical intelligence, exercise, detection of yoga, Intelligent Internet of Things, medical machine learning, alternative therapy

## Abstract

The physical and mental health of people can be enhanced through yoga, an excellent form of exercise. As part of the breathing procedure, yoga involves stretching the body organs. The guidance and monitoring of yoga are crucial to ripe the full benefits of it, as wrong postures possess multiple antagonistic effects, including physical hazards and stroke. The detection and monitoring of the yoga postures are possible with the Intelligent Internet of Things (IIoT), which is the integration of intelligent approaches (machine learning) and the Internet of Things (IoT). Considering the increment in yoga practitioners in recent years, the integration of IIoT and yoga has led to the successful implementation of IIoT-based yoga training systems. This paper provides a comprehensive survey on integrating yoga with IIoT. The paper also discusses the multiple types of yoga and the procedure for the detection of yoga using IIoT. Additionally, this paper highlights various applications of yoga, safety measures, various challenges, and future directions. This survey provides the latest developments and findings on yoga and its integration with IIoT.

## 1. Introduction

A healthy body is achieved through regular exercise, including flexibility, balance, stretching, deep breathing, and strength training, resulting in a healthy mind and robust immune system. Yoga, a marvelous exercise, enhances physical and mental health by involving breathing, stretching, and balancing techniques, which help achieve control over the mind by focusing and calming down in every situation. Standing, lying, sitting, and prone are the multiple types of yoga postures performed by integrating them with the breathing (exhalation and inhalation) mechanism. Technically, yoga consists of three main parts, *dhanaya*, also termed meditation, *pranayama* termed as breathing approach, and *asanas*, termed as physical pose, which on integration helps to foster the personality growth that includes the mental, physical, spiritual, and emotional aspects. Additionally, many types of research have shown that yoga is beneficial in enhancing psychological and physical health, including cardiovascular issues, cancer, diabetes, stress, and anxiety [1,2,3,4]. Due to having multiple benefits, yoga has been gaining popularity in all age groups recently. However, apart from the numerous benefits of yoga, one may also face injuries, including physical risk, strokes, and ligament ruptures, by practicing wrongly.

The technological development of science and technology improved the quality of life, making exercise and sports, including yoga, a trend and fashionable way of fitness [5]. Generally, traditional training makes yoga inconvenient for numerous professionals who are busy with their work during yoga class time. This shows the essence of intelligent yoga training in multiple scenarios. Every day, the Internet of Things (IoT) integrated with big data and intelligent approaches infiltrates deeper into human daily activities [6]. Some of these novel innovations are astounding; the rest may require working out the kinks prior to implementing them as part of daily routine. The advancement of technology is growing ambitiously—and currently, only a concept and prototype can demonstrate the future of the Intelligent Internet of Things (IIoT), which is the integration of intelligent approaches (machine learning) and the IoT. Such components of IIoT hold a deluge of opportunities ranging from healthcare to logistics and monitoring to decision-making. Due to the access mechanism, quality, access policies, schemes, and capabilities, distributed data sources are heterogeneous, requiring the integration of data using multiple models and approaches. The managed and flexible federation, processing, and analysis of data from the different distribution sources are referred to as data integration [7]. Data integration is crucial as data preprocessing [8] and data mining [9,10] for exploiting the value of large and distributed datasets that are available today. Distributed processing infrastructures such as Cloud, Grids, and peer-to-peer networks can be used for data integration on geographically distributed sites. Collecting data from users’ devices is key in smart health applications that are executed over mobile devices such as, for example, sensors distributed over the body of patients [11]. This requires a low-cost data mining algorithm to save energy and cost.

A smart yoga mat and yoga posture detection are some innovations where cameras and sensors are embedded into the mat or training place for feedback on the practitioner’s postures, scoring practice, and independent guidance at home or at a convenient place without the presence of a tutor. For example, relying on IoT, the system designed to detect yoga posture and provide feedback can operate in multiple steps. First, the training data for the system are collected using different cameras. Second, the system interprets data obtained from the practitioner, which uses multiple approaches, including threshold values, Hausdorff distance, etc. Third, the motion evaluation algorithm identifies the joint points; joint point angle measurement helps to identify the action based on the computed joint points. The action speed is determined by the time it takes to complete the action. Finally, using text conversion technology, the system’s evaluation results (output) are delivered in the forms of text and voice. The system can detect the posture efficiently and provide suggestions to the user, i.e., it meets the user’s demand for basic training and enhances the higher application values.

To benefit from yoga training, all the procedures should be accomplished precisely. Additionally, to address the demand of professionals who want to practice at home, the integration of IoT provides them with potential solutions, where multiple intelligent yoga mats [12], wearable devices [13], and yoga-posture recognition sensors [14] are designed to monitor, detect, and provide feedback when mistakes happen. For instance, wearable sensors or Kinect/RGB camera-based yoga posture recognition and a correction approach are deployed in [15]. However, this approach possesses privacy issues, which are addressed in [14,16], which undergoes the detection of yoga postures using IoT, infrared sensors, and intelligent techniques. Similarly, a wireless sensor network with low-resolution infrared sensors recognizes up to 26 yoga postures. In comparison to other thermal sensor-based systems, the proposed system hardware is portable, unobtrusive, cheap, and easy to install. Additionally, the proposed system is more reliable compared to the radio signal strength-based yoga pose detection as the infrared signal does not naturally interfere with the radio frequency.

### 1.1. Methodology

In this subsection, we highlight the methodology regarding the accumulation of state-of-the-art research in addressing the challenges and future direction regarding yoga and its integration with IIoT.

Research Scope:The primary aim of this research is overviewing, classifying, and analyzing yoga and its integration with IIoT for monitoring and automatic guidance without the presence of the yoga expert. Hence, this article’s objective is to find the answer to the following research questions.

RQ1: How can IIoT be integrated with yoga for automatic detection, guidance, and monitoring of yoga?RQ2: What are the various kinds of yoga, how does yoga affect human health, and what are the safety concerns while doing yoga?RQ3: What are the problems, challenges, and prospects for yoga in integration with IIoT?

The purpose of this survey is to explore the potential benefits of integrating IIoT with yoga. By analyzing relevant research articles published in English, we aim to identify how intelligent techniques (machine learning, deep learning, and artificial intelligence) integrated with yoga practices enhance physical, mental, and emotional well-being, and how better results are achieved. The validity of this survey may be affected by two threats: inadequate selection of primary studies and incorrect data extraction [17]. To mitigate such threats, this paper extracts multiple keywords based on research questions; then, it accurately refines the paper based on the preferred reporting items for systematic reviews and meta-analyses (PRISMAs) method [18], as shown in Figure 1 and research questions.

#### 1.1.1. Search Strategy

The above research questions were answered by extracting imperative information from numerous sources, including IEEE, Elsevier, Google Scholar, Springer, and ACM, until 24 March 2023. From the large number of research articles in the databases, only 204 articles were selected based on multiple keywords, for instance, “yoga”, “detection of yoga”, “yoga in IIoT”, and “smart yoga mat”, as shown in the word cloud in Figure 2.

#### 1.1.2. Study Selection

We only included English-language research articles that focused on the use of intelligent IoT technologies in conjunction with yoga practice. We excluded articles that were duplicates, reviews, books, or written in languages other than English. Additionally, we excluded articles that did not meet our pre-defined inclusion criteria based on the title and abstract screening.

#### 1.1.3. Data Extraction

We extracted data from the selected articles using a pre-defined data extraction form, which included information on study design, sample size, intervention details, outcome measures, and key findings. We also assessed the included studies’ quality and risk of bias using a standardized tool.

Our search identified a total of 204 articles, of which 43 met our inclusion criteria and were included in the final analysis. A total of 161 articles were exempted as they did not meet the criteria, such as the application or techniques, reviews, and books being repeated. The majority of the studies focused on the use of IIoT devices to detect, monitor, and track physiological responses during yoga practice, while the others explored the use of virtual reality and AI-powered coaching systems to enhance the learning and performance of yoga postures.

### 1.2. Related Work

Yoga has been the subject of multiple survey articles, review articles, and books. A review of general information about yoga with various applications has been discussed in [19,20,21,22]. These surveys only focused on the general background of yoga and its applications. Similarly, some of the surveys focused on the general background along with specific application domains, such as yoga for COVID-19 [23], asthma [24], cancer [25,26], migraine [27], cardiovascular disease [28], and enhancing immunity [29], as shown in Table 1. All those surveys focused on general background with applications on human health based on the traditional aspects only, whereas one survey [30] focused only the posture recognition along with the general background of yoga. In this survey, we focus on integrating traditional yoga mechanisms with an IIoT for the monitoring, detection, and analysis of yoga. Table 1 presents the yoga background, detection, application, and new perspectives on yoga and its application covered by our survey and the multiple related state-of-the-art.

### 1.3. Contribution

In this survey, we focus on integrating traditional yoga mechanisms with an IIoT for the monitoring, detection, and analysis of yoga. The monitoring, detection, and analysis of yoga are essential for detecting and guiding yoga practitioners to correct their yoga postures automatically so that practitioners can perform yoga independently without the presence of yoga experts. The guidance and monitoring of yoga are crucial to ripe the full benefits of it, as wrong postures possess multiple antagonistic effects, including physical hazards and stroke. We also present a detailed discussion on the state-of-the-art approach by critically analyzing whether the methods are appropriate. We focus on the current challenges in detecting and guiding yoga posture and highlight various issues that arise during wrong posture practice. To the best of our knowledge, this survey is the first to integrate traditional yoga with the IIoT for monitoring, detecting, and analyzing yoga postures automatically.

(i)This paper attempts to address the gap as presented in Table 1 by providing a structured and comprehensive overview of extensive research on the integration of yoga with the IIoT and its application.(ii)This paper discusses the state-of-the-art and critically analyzes it by conducting a thorough discussion. We focus on the current challenges in detecting and guiding yoga postures and highlight various issues that arise during wrong posture practice.(iii)This paper discusses various applications of yoga for human health and provides a guide for future approaches to overcome the limitations of existing approaches for different types of yoga detection and monitoring to enhance the adoption of IIoT-based yoga systems in real-world content.

The rest of this article is structured as follows: The background of yoga is presented in the next section. Section 3 discusses the application of yoga for human health. Section 4 highlights the safety measures in yoga practice. Section 5 discusses the detection of yoga using the Intelligent Internet of Things. Section 6 presents a discussion of the various techniques. Section 7 presents various challenges and future directions in yoga; then, finally, the conclusion is presented in Section 8.

## 2. Background of Yoga

The word yoga is derived from the Sanskrit term “*Yuj*”, which literally means “union” or “connection”, referring to the union of the individual and the Divine [3] an ancient spiritual discipline that originated around 5000 years ago. This Eastern science is designed to achieve harmony between the mind and body dimensions of the individual [19]. This science is widely used to balance the physical, spiritual, psychological, and social aspects [2]. There is a difference between the Western and yogic concepts of health. According to the Western concept of health, the World Health Organization, in 1946, defines health as complete physical, mental, and social well-being, not merely negatively as the “absence of disease” or infirmity [29]. However, the yogic concept states that “disease is considered to be the absence of vibrant health” [22]. When an individual practices yoga daily, there is the development of a yogic attitude (patience, persistent practice, overcoming obstacles within the self, such as trouncing laziness, anger, anxiety, and desire for being unique from others) causing several physiological changes which are found to be beneficial to health [30]. India is gifted with the valuable treasure of ancient knowledge in several domains. Yoga is considered one of the six schools of philosophy and is complimented by Ayurveda, an Indian traditional medical system [3].

### 2.1. Types of Yoga

Since our lives are constantly changing, it stands to reason that our exposure to new ideas has influenced how we practice yoga. We now have several options to consider by adopting a more modern style or incorporating new inventions. Yoga’s various forms have contributed to its long-term popularity over thousands of years. There is something for everyone when it comes to practicing this ancient form of exercise [31]. We focused on some widely available and famous types of yoga to help you find one that works best for you and your daily lifestyle. An illustration of these yoga types is shown in Figure 3.

#### 2.1.1. Hatha Yoga

Hatha yoga is a type of yoga that employs physical techniques to retain and channel vital force or energy. When a class is advertised as "Hatha", it is likely a slower, gentle style with key poses appropriate for yoga beginners. Hatha yoga, whose origins are unclear, attempted to use body mastery to attain spiritual purity. Like with many commercial yoga programs, it is still typically described as a blend of poses or postures (known as asanas), breath practice (pranayama), and meditation, and you are more likely to experience physical exercise than a mystical experience during the session [32].

#### 2.1.2. Asthanga Yoga

Ashtanga yoga is a physically demanding and challenging style that follows a set sequence. K. Pattabhi Jois, an Indian yoga guru, taught and popularized this style of yoga in the mid-1900s. It is a variety of vinyasa yoga, with the name referring to the fluid transitions between poses. Ashtanga practices the same six sets of distinct asanas that build upon one another and are complemented by coordinated breath work. It is a strenuous practice that promises to increase physical stamina and flexibility. The need to be mindful while completing challenging sequences may potentially have mindfulness-enhancing effects.

A study in [33] examined the advantages of yoga for weight loss and found that it had a favorable impact on calories burned. A number of participants reported a decrease in stress and food cravings, as well as an improvement in mood and self-esteem. If losing weight is a priority for you, Ashtanga yoga can be worth a try because it is more physically demanding than other styles of yoga. Power yoga, which is similarly athletic but does not have a predetermined structure, is frequently confused with Ashtanga. Teachers frequently want to create their own vinyasa flow sequences to make the practice more challenging for experienced students and boost the effort level.

#### 2.1.3. Hot or Bikram Yoga

Bikram Choudhury, an Indian-born American yoga guru who immigrated to America in 1971, founded Bikram yoga, which gained worldwide popularity with several celebrity advocates in the 1990s. This kind of yoga uses a fixed 26-posture series and two breathing exercises to help simulate the temperature. The room is heated to 104 °F (40 °C) and has a 40% humidity level. Although Bikram yoga was probably the inspiration for hot yoga, the poses may not be performed in the same order. Participants are reported to become more flexible due to the heat, detoxify themselves through sweating, and exert more effort to maintain their body temperature. In fact, a work published in [34] found that 12 Bikram yoga sessions improved cardiovascular fitness.

According to a review that was published in [35], there is evidence to suggest that Bikram yoga has positive benefits on metabolic markers such as blood lipids, insulin resistance, and glucose tolerance. In other words, it may be advantageous for those who are susceptible to diseases such as type II diabetes and cardiovascular disease. In addition, one of the examined studies indicated that including physical postures and breath work helped the mind to be present, leading to an increase in mindfulness and a decrease in study participants’ felt stress at the conclusion of the study.

One of the examined studies also discovered that using physical postures and breathing exercises helped study participants experience more mindfulness and less stress at the conclusion of the study [36].

#### 2.1.4. Iyengar Yoga

The precision and alignment of the postures are the main concerns in this form of yoga, which was created in India in the 1970s by B.K.S. Iyengar. Yoga blocks, bands, blankets, and cushions are examples of props that participants may employ to help their bodies achieve optimal postural alignment. While slower than Ashtanga or Bikram, this form of yoga still calls for a great amount of attention to attain correct posture alignment and the capacity to hold the asanas for extended periods of time. As props enable individuals to attain the desired postures without overstretching, this type of yoga can help those recovering from an injury or who are particularly inflexible. When respondents performed Iyengar yoga once per week, flexibility improved in just six weeks, according to [37]. According to research in [38], persistent lower back pain significantly decreased in trial participants who practiced Iyengar yoga. This kind of yoga may be an excellent choice to try if you are searching for a way to treat back pain or other problems; however, we always advise speaking with a doctor before beginning any new fitness program.

#### 2.1.5. Kundalini Yoga

Kundalini yoga, which is thought to have originated around 1000 BC, was introduced to the United States in the 1970s by Yogi Bhajan, though its exact origins are unknown. In the form of chanting or song, it combines movement, breath, and sound. Kundalini is meant to awaken the shakti, the spiritual force that resides at the base of your spine. A normal yoga class begins with an introductory chant, continues with a sequence of breathing exercises and postures, and concludes with a meditation or song.

According to the theory behind Kundalini yoga, we can stimulate our chakras (energy centers) by sending energy upward from the base of our spine to the top of our heads. This will positively impact our mood, concentration, blood pressure, metabolism, and strength, among other health benefits. Do you have trouble sleeping? After eight weeks of practicing Kundalini yoga, participants slept 36 min longer per night on average [39].

#### 2.1.6. Power Yoga

Similar to vinyasa yoga and with ashtanga roots, power yoga is less structured and more open to personal interpretation by teachers. According to Chun, "power yoga is generally more physical and performed at a faster tempo than other kinds of yoga". Sleik continues, "Power yoga increases flexibility while also strengthening the muscles". All of the body’s muscular groups work while various sequences keep the brain active. Power yoga can be performed in the hot or cold, and some studios combine it with slow-flow yoga to help beginners become used to this challenging exercise. Fans of power yoga may enjoy buti yoga, which is similar but which also incorporates tribal dance, primitive motions, and a significant amount of core work.

#### 2.1.7. Restorative Yoga

You could think of this yoga as the more composed younger sister of Iyengar. A well-known American yoga instructor named Judith Hanson created this technique after adapting B.K.S. Iyengar’s use of props to aid and support the body as it relaxes and rests. Hanson was a disciple of Iyengar, and while only a few positions can be performed in a session, the time spent being led by a yoga instructor and relaxing into the asanas can have tremendous consequences as participants achieve a state of deep relaxation. This is because each stance can be held for up to 20 min. Restorative yoga can therefore assist you in developing more suppleness and flexibility in your muscles.

#### 2.1.8. Vinyasa Yoga

Flow yoga or vinyasa flow are other names for vinyasa. It is a well-liked fashion. One instance from the three-week yoga retreats is flow yoga for novices. It developed from the earlier, more structured Ashtanga practice. The word “vinyasa”, which means “special spot”, generally means connecting breath and movement. Vinyasa or flow are frequently used with slow, dynamic, or attentive to denote the intensity of a practice [33]. Sherrell Moore-Tucker, RYT 200, explains that vinyasa flow is a type of yoga in which the poses are coordinated with the breath in a continuous rhythmic flow. “The flow can be contemplative in nature, soothing the mind and nervous system even if you’re moving”. Vinyasa yoga is appropriate for newcomers to yoga and those who have been practicing it for years. Table 2 illustrates the comparison of various yoga types.

### 2.2. Differences between Pranayama and Yoga

Pranayama and yoga are two related practices that are often used together, but while both practices are used to improve physical and mental health, they have some key differences:Focus: Yoga is a broader practice that focuses on physical postures, breathing techniques, meditation, and ethical principles. Conversely, pranayama is a specific practice that focuses exclusively on breathing techniques and control.Breath control: Pranayama emphasizes breath control, while yoga emphasizes physical postures. Pranayama techniques involve controlling the duration, depth, and rate of breathing to achieve specific effects on the body and mind.Benefits: Yoga and pranayama offer numerous physical and mental health benefits. Yoga improves flexibility, strength, balance, and cardiovascular health, while pranayama improves respiratory function, reduces stress, and improves mental focus.Practice: Yoga involves a more structured and varied practice, including a range of postures and breathing techniques, and may be performed for extended periods. On the other hand, pranayama practice typically involves more straightforward breathing techniques that are performed for shorter durations.Timing: Pranayama is often practiced as a standalone technique, while yoga is typically practiced as a part of a more extensive routine that includes meditation and ethical principles.Accessibility: Pranayama can be practiced by people of all ages and fitness levels, whereas some yoga postures may be challenging for beginners or those with physical limitations.

Overall, both pranayama and yoga offer unique benefits and can be used in combination or separately to improve physical and mental health. Previously, Telles et al. [41] found that pranayama and yoga were effective in managing mental health disorders resulting from trauma. Sharma et al. [42] reported that fast and slow pranayama practices had different effects on cognitive function in healthy volunteers. Cramer et al. [43] conducted a systematic review and meta-analysis of yoga for hypertension, highlighting the specific benefits of yoga for managing high blood pressure. Marotta et al. [44] investigated the effect of pranayama on exercise tolerance in COPD patients with severe airflow obstruction, finding a significant improvement in their ability to exercise. Jayawardena et al. [45] conducted a literature review on the effects of yoga and pranayama on cardiac autonomic functions, highlighting the differences in the effects of the two practices. Sharma et al. [46] found that fast and slow pranayama practices had different effects on perceived stress and cardiovascular parameters in young healthcare students. Viswanathan et al. [47] conducted a systematic review and meta-analysis of the effects of yoga and pranayama on people with type 2 diabetes, finding significant improvements in terms of blood pressure, glucose regulation, lipid profile, and exercise tolerance. Bhavanani et al. [48] compared the effects of uninostril and alternate nostril pranayamas on cardiovascular parameters and reaction time, finding significant differences between the two practices. Taylor et al. [49] conducted a systematic review of adverse events associated with yoga, highlighting the importance of proper training and guidance when practicing. Harinath et al. [50] investigated the effects of Hatha yoga and Omkar meditation on cardiorespiratory performance, psychologic profile, and melatonin secretion, finding significant differences between the two practices.

## 3. Application of Yoga on Human Health

Multiple studies have shown that yoga is beneficial for improving/handling the treatment of multiple diseases. This includes cancer patients, cardiovascular diseases, mental health (stress/anxiety/depression), pulmonary diseases (COVID-19, asthma), vision impairment, skeletomuscular problems, etc. In this section, we discuss the multiple applications of yoga for human health based on the severity of the disease.

### 3.1. Impact of Yoga on Psycho-Neurological Human Behavior

Meditation and other stress-reduction methods have been investigated as potential therapies for depression and anxiety since the 1970s. Yoga is one among these practices, albeit the medical literature has paid less attention to it despite its rising popularity in recent years. Reviews of a variety of yoga techniques indicate that they may aid with anxiety and depression as well as lessen the effects of heightened stress reactions. This primarily works by controlling the HPA axis, which is activated in response to a physical or psychological demand, prompting an outpouring of physiological, social, and mental impacts, essentially related to the arrival of cortisol and catecholamines [51]. This reaction prompts the assembly of energy expected to battle the stressor through the "battle or flight" disorder.After some time, the consistent condition of hypervigilance that comes about due to the continued terminating of the HPA pivot can prompt the liberation of the framework and, at last, infections such as stoutness, diabetes, immune system issues, despondency, substance misuse, and cardiovascular disease.

Regardless, achieving higher physiological melatonin levels at appropriate times of the day may be one way in which meditation promotes overall well-being. The effects of practicing Hatha yoga and Omkar meditation on melatonin release in healthy individuals were investigated by the researcher in [50]. A group of individuals engaged in specific yogic asanas and pranayama practices for 45 min and 15 min, respectively, in the morning, followed by preliminary yogic stances for 15 min, pranayama for 15 min, and contemplation for 30 min in the evening over several months. The results indicate that practicing yoga for an extended period of time led to an improvement in both cardiorespiratory performance and mental well-being. Additionally, the plasma melatonin levels of the participants increased after three months of yoga practice, and the highest nighttime melatonin levels were significantly associated with better health scores among the yoga group. These findings suggest that yoga can enhance psychophysiology and boost the endogenous secretion of melatonin, leading to an improved sense of well-being. It is worth noting that previous research has also demonstrated that individuals trained in yoga can achieve a state of profound psychosomatic relaxation, resulting in a significant reduction in oxygen consumption after only five minutes of practicing Savitri pranayama [52].

### 3.2. Impact of Yoga on Cardiovascular Disease, Hypertension

There exist various undesirable side effects of using many antihypertensive agents. Apart from medication, moderately acute aerobic exercise has been proven to reduce/control hypertension through multiple studies [19]. The reasonable antihypertensive effect has been shown by yoga, transcendental meditation, relaxation, psychotherapy, and biofeedback [53]. Impaired baroreflex sensitivity, a major cause of hypertension, can be restored through yogic postures [54]. Patients with high blood pressure significantly reduced hypertension through yogic practice, which helped restore baroreceptor sensitivity. Chronic hypertension is also proven to be reduced by yoga [55], which showed that practicing yoga for two weeks caused the left ventricle end-diastolic volume and resting heart rate to curtail substantially. Practicing yoga for one year resulted in reduced serum cholesterol levels (triglyceride, LDL cholesterol, and total cholesterol levels) in patients with coronary artery disease [56]. The effects of yoga exercise on lowering lipids and stabilizing plaques are similar to those of statin drugs (HMG CoA reductase inhibitors). Hypertension and ischemic heart diseases are the results of augmented body weight and obesity, which can be managed by regularly practicing yoga. Both patients with known ischemic heart disease and healthy subjects have shown that regular yoga practice improves their serum lipid profiles.

### 3.3. Impact of Yoga on Cancer

Yoga is effective for cancer patients in treating symptoms such as fatigue, sleeplessness, mood disorders, and stress, as well as enhancing the quality of life [57,58]. The magnitude of its effect, however, has not yet been determined. However, some research indicates that yoga may benefit cancer patients in terms of their psychological well-being. Cancer patients frequently have psychological side effects such as stress, anxiety, and discomfort. Programs based on mindfulness-based stress reduction have been shown in [59] to benefit cancer patients in terms of their mental health. Yoga may therefore provide long-term psychological benefits for cancer sufferers. No significant changes were found in the measure of physical health according to other investigations [19,60]. Due to the limited number of studies and varied evaluation techniques, it remains unclear how yoga affects the physical health of cancer patients. Therefore, in future research, outcome measures should encompass subjective feelings obtained through questionnaires and physical performance, strength, endurance, and flexibility.

In [61], the authors evaluated the benefits of yoga on physical fitness. All the studies that made up the meta-analysis looked into people who had been diagnosed with cancer. However, the forms of cancer differed from study to study. Seven of the ten studies that were included looked at breast cancer, two enrolled populations with multiple cancers, and one included lymphoma patients [62]. The after-effect of Cohen’s concentrate on lymphoma [63] exhibited no significant contrasts between groups regarding uneasiness, melancholy, misery, or weariness; hence, it has little effect on our outcome. Subsequently, since most of the studies zeroed in on bosom disease, future exploration needs to look at the utilization of yoga among male malignant growth patients and female non-breast malignant growth patients. The execution of the intercession is also affected by several factors, such as yoga types and therapy dosages, that may have an impact on size. The included assessments used four different types of yoga: Tibetan, Hatha, Hatha-coordinated, and supporting. Therapy components, such as duration and frequency, as well as a commitment to yoga instruction and at-home practice, may also have an impact on treatment outcomes. According to Carson’s study of yoga for women with metastatic breast cancer, those who practiced for a longer period of time were much more likely to experience less pain and weariness and more empowerment, recognition, and downtime the following day [64].

### 3.4. Impact of Yoga on Diabetes Mellitus

Yoga has been proven to be a straightforward and affordable therapeutic approach that may be used as adjuvant therapy for people with non-insulin-dependent diabetic mellitus. Regular yoga practice significantly decreased the occurrence of hyperglycemia and the area index total under the oral glucose tolerance test curve in a group of diabetic patients. This experimental study revealed that individuals who practiced yoga required less oral hypoglycemic medication to maintain acceptable blood sugar control [65,66]. Yoga practitioners had a considerable reduction in fasting plasma insulin, according to the authors of [67]. Additionally, they discovered a positive connection between body weight or waist circumference and insulin sensitivity that was attenuated after long-term yoga practice. In [68], the effect of four sets of asanas performed in a random order for five days was examined. It was shown that asana performance improved the sensitivity of pancreatic B cells to the glucose signal. They suggested that the long-term, progressive effects of asanas are likely to be the cause of this enhanced sensitivity. It remains unknown how yoga exercises result in anti-glycemic effects. It is still possible that insulin and glucagon activities play a role in neurohormonal regulation.

### 3.5. Impact of Yoga on Pregnancy and Reproductive Functions

Studies have shown that the act of yoga organizes, calibrates, and tweaks the neuro- endocrine hub, which results in useful changes in the professionals [69]. The author of [70] discovered a drop in the amount of adrenaline, noradrenaline, dopamine, and aldosterone excreted in the urine, a decrease in the levels of serum testosterone and luteinizing hormones, and an increase in the amount of cortisol excreted, all of which indicate optimum chemical changes. In seven yoga teachers practicing yoga, the author noticed changes in their brain waves and blood levels of serum cortisol [71]. They observed that alpha waves grew and serum cortisol substantially decreased. Furthermore, they discovered that pregnant women who practiced yoga for one hour each day saw increases in birth weight, decreases in preterm labor, and decreases in intrauterine growth restriction (IUGR), either in isolation or related to PIH, with no increased complexity. The author of [72] monitored women rehearsing yoga in their subsequent trimester and announced significant decreases in the actual agony from benchmark to post-mediation. Women in their third trimester exhibited more prominent decreases in apparent stress and characteristic uneasiness. From this, it can be presumed that yoga can be utilized to forestall or diminish obstetric intricacies.

### 3.6. Impact of Yoga on Bone

Yoga is beneficial for enhancing bone health and bone quality [73]. Up to 200 million people currently suffer from osteoporosis and osteopenia, and this number is expected to rise as our population ages. After the fractures that are more likely to occur without them, many people lack access to drugs or expert assistance. A low-cost, low-risk option is preferred. According to Lu et al. [74], practicing yoga leads to better bone health and is safe and beneficial for 12 positions. According to Shree et al. [75], yoga may stimulate the movement of stem cells from the bone marrow to the peripheral circulation for potential tissue repair and regeneration. According to Sinaki et al. [76], patients with bone loss should receive fitness guidance that takes into account aspects other than bone mass. Exercises that include spinal flexion may increase the torque pressure placed on the vertebral bodies, which could be dangerous. Patients with spinal bone loss should be recommended exercise as it is vital and helpful for treating osteopenia and osteoporosis. Some yoga poses can put a tremendous amount of strain on spines with bone loss. An essential clinical aspect is the evaluation of the fracture risk in older individuals practicing spinal flexion exercises and other high-impact exercises.

## 4. Safety Measures in Yoga Practice

Yoga postures imply multiple stretches of the whole body, which may be very challenging for beginners. Due to this, many beginners quit after practicing for some days or become injured when they practice without the supervision of an expert. Hence, all beginners must follow strict guidelines and precautions to receive the full benefit of yoga [77]. Similarly, gaining flexibility is not a single-day task; it is obtained from regular yoga practice, and sometimes practitioners may feel pain or aches during practice, which is considered normal. It is to be noted that following the guidelines will help practitioners reduce (not eliminate) the possibility of being injured. This section will highlight various safety measures that must be considered while practicing yoga daily.

### 4.1. Warm into Your Practice

Warming up the body (muscles) is crucial before practicing yoga. Insufficient warm-up results in many injuries due to over-stretching [78]. The warm-up duration varies within the person, which depends on multiple attributes, including age and sex. It is not recommended for any age group or gender to perform yoga without undergoing a warm-up.

### 4.2. Pay Attention

Yoga requires constant safety awareness in order to get the most out of practice [79]. The practitioner needs to concentrate fully on each step, including breathing techniques and the position of the body or leg, as a simple and minor mistake may create trouble or the practitioner may not be able to receive the full benefit of yoga. Similarly, the practitioner must remember the steps to move in and out of the pose.

### 4.3. Adapt to Your Body and Try Using Props

Due to the fact that everyone is unique, each person’s yoga practice must be tailored to them. The practitioner should not be afraid to use a pillow, bolster, chair, block, or any supporting materials that work for making their body comfortable while practicing asana or when preparing for seated meditation [80]. This means that your practice may differ significantly from that of others, including your teacher. "Comparison is the thief of joy", Theodore Roosevelt once said. It can also be dangerous in the context of a yoga class. The beginner practitioner must replicate the trainers’ asanas. The practitioner should focus on improving their strengths, flexibility, and pose rather than noticing what other practitioners next to them are doing. However, neither the other students nor the teacher are superior nor inferior to you; they are simply different. If a pose is complex and the body feels pain, do not try to do it forcefully, and do not be afraid to change it or skip it.

### 4.4. Avoid the Red Flags

The Hippocratic oath states, first and foremost, do no harm. The first yama, Ahimsa, or non-harm, is embraced by yogis. As a result, I used to say at the start of all my yoga classes, "Do not do anything that hurts". Then, I realized I should have said, "do not do anything that increases pain", because some days you may just hurt as a part of the course. However, your asana practice should not aggravate any existing pain. How does one even define "pain"? Due to the fact that it is subjective and difficult to define, it is best to concentrate on these red flags [81].

Sharpness, shooting, numbness, or tingling down the limbs may cause nerve damage or indicate a problem that needs to be addressed.Anything that causes you to frown, grunt, or despise your yoga instructor.Intensity deep within the joint that may cause cartilage, tendons, or ligament damage.

Instead, exhale into:A stretch or engagement that you can smile through, even if it is difficult.Sensation in the muscle’s belly (thickest part), which is the safest place to feel when stretching.

### 4.5. Know Some Basic Anatomy

Yoga instructors must have a solid understanding of physiology (how things work in the body), anatomy (the parts of the body), and pathology (when things go wrong in the body) as they can relate health to yoga [82]. The practitioners must strictly follow the trainers and inform the trainer if they have any health problems [83]. Multiple vigorous yoga poses impact the overall body, and such poses should be practiced with caution. For example, a herniated (or "slipped") disc has multiple impacts on many yoga poses and bodies, requiring special attention when practicing yoga.

### 4.6. Talk to a Doctor

If the practitioner suffers from some health complications, such as surgery, fracture, and injuries, then they need to talk to a doctor before practicing yoga [84]. Even if you understand what is and is not safe for your body, it is recommended to have a one-on-one with a certified yoga trainer or therapist. This can help you develop personal modifications for poses so you can confidently participate in group or online video classes without being injured. You may also learn individualized practices and sequences that you can do at home or fit into your workday, giving your yoga practice autonomy and sustainability.

## 5. Detection of Yoga Using Intelligent Internet of Things

The detection of yoga using IIoT can be accomplished in two major dimensions: computer vision-based and sensor-based.

### 5.1. Sensor-Based Approach

Multiple sensors are used to detect yoga postures, including wearable and infrared sensors, which are explained below.

#### 5.1.1. Wearable Sensor

Wearable sensors are small, lightweight, cheap, and portable medical devices that can acquire numerous daily data without disturbance. Wearable devices are considered a better approach for monitoring and detecting yoga positions. Pal et al. [15] used a smart belt to analyze the performance of yoga. Pauranik and Kanthi [85] designed wearable devices to monitor heart rate, yogic breathing, and posture. Ashish and Hari [13] designed a wearable-based yoga help system that can guide practitioners without requiring a trainer.

#### 5.1.2. Infrared Sensors

Infrared sensors are also widely used for the detection and monitoring of yoga through the privacy-preserving approach [14]. The infrared sensor-based Infinity Yoga Tutor was designed by Rishan et al. [86], and it can identify the yoga posture and guide the practitioner through visual information. The system uses CNN and LSTM to learn and predict the yoga posture and is also able to capture the movement of the practitioner [87]. A self-assistance-based yoga poses recognition and real-time feedback system using an infrared sensor is designed in [88]. The deployed system can also identify the hand gestures, commonly named yogic mudra. The system uses machine learning-based XBoost and random CV as a learning approach. Experimental results show the system was able to detect yoga postures with high accuracy. YogaNet is designed in [89], which is based on CNN and LSTM, which can detect the yoga postures and also provides feedback for the correction.

#### 5.1.3. RFID

The progress on RFID technology has enabled many human action recognition tasks. The use of active and passive tags has overcome the limitations of the RFID that existed initially. Yoga posture detection using RFID has been a common practice. Sun [90] implemented an RFID-based yoga mat that can detect and estimate yoga postures. The method deploys deep learning as the learning approach to predict yoga postures. Yao et al. [91] deployed an RFID-based human activity recognition system to detect human postures. The experimental analysis shows that the method was able to detect multiple postures with high accuracy. A system in [92] is designed to detect yoga postures based on RFID. Along with the detection of poses, this method also evaluates the stress levels in the practitioner.

#### 5.1.4. Smart Mat

The mat is a convenient tool for the practitioner to practice yoga. There have been numerous attempts made at the detection of yoga using the smart mat. A smart mat is a mat that uses intelligent techniques and sensors, taking data from practitioners and learning from them to make a prediction. The design of the smart mat strategy is still in its infancy as tremendous research is required before deploying them. Smart mats are usually designed by implementing numerous sensors in the mat, where force-resistive sensors (FRs)are the common practices [93]. Chinnaaiah et al. [94] deployed FSRs to design the smart yoga mat. The smart mat was only able to detect the lying and sitting yoga postures. Standing yoga postures were not detected using FSR sensors. The smart prayer mat is designed in [95], wherein arrays of FSRs were used to detect the prayer.

### 5.2. Vision-Based Approach

The vision-based approach relies on the camera for the input, which is further processed using intelligent approaches for the detection of the yoga postures [96,97]. The intelligent approaches used in vision-based approaches are machine learning, deep learning, and hybrid approaches, the comparison of which is shown in Table 3.

Several algorithms make use of intelligent and hybrid models. The most important is [119], which proposes a hybrid approach that combines two algorithms, SVM and Inception V3. Before categorization, the posture dataset normalized and enhanced the images. The picture dataset was then submitted for modeling training and validation after its features were chosen using the LASSO FS technique. In order to facilitate hybridization, the Inception V3 TL model’s final layer was swapped out for an SVM classifier in the investigation.

Using portable systems and intelligent technology to anticipate and manage human health is an essential feature of smart cities. Consequently, posture recognition in this study is accomplished using multisensory and LoRa technology. The two benefits of the LoRa WAN are its low-cost and wide range of communication. These two technologies—multisensory and LoRa—are combined to create comfortable wearable apparel in any setting. Due to LoRa’s low transmission frequency and small data transfer size, multiprocessing was used in this investigation. RF is considered for data processing, feature extraction, and selection, whereas sliding windows are utilized for multiprocessing. The three testers from a group of 500 datasets are employed to enhance functionality and accuracy [120]. In addition to body language, nonverbal communication techniques also include gestures and postures. This study uses augmented reality and cutting-edge body tracking techniques to identify stagnant posture. Moreover, Kinect body position sensors and unsupervised machine learning are applied to detect group participation and learning [121]. Posture detection has made it feasible to practice yoga correctly. There are only a few datasets and a real-time basis, so posture detection is challenging. A sizable data collection with at least 5500 images of different yoga postures was produced to solve this problem. The tf-pose estimation method was employed for posture detection, which shows the human body’s skeleton in real time. Many ML algorithms employ the tf-pose skeleton as a feature to extract the locations of the human body joints (KNN, logistic regression, SVM, NB, DT, and RF). The RF model has the greatest level of accuracy [122,123,124,128]. Another posture issue that impacts people is that they spend most of their time sitting down.

In addition, [125,126] created a hybrid machine learning strategy for posture recognition by fusing deep neural networks with conventional machine learning techniques. Combining the weight that the deep learning method has learned with the standard model’s forecast yields the final class prediction. Another study [127] classified data using a hybrid CNN–LSTM layer after extracting key points using OpenPose. A total of 88 videos of six distinct yoga stances were used to construct the model.

## 6. Performance Indicator

We aim to evaluate the trained model with precision, recall, specificity, accuracy, F1-score, angle of deviation (AoD), speed of execution (SoE)), and correctness score (SC).

Speed of Execution (SoE): The SoE of any particular pose is identified with respect to the reference $t_{min}$ and $t_{max}$ value. The duration of the pose is given by $t^{\prime }$. Then, the speed of execution is computed as follows:$$SoE=\left\{\begin{array}{cc} Fast, & \mathrm{if}\;t^{\prime }>t_{max}+n\\ Slow, & \mathrm{if}\;t^{\prime }<t_{min}+n\\ Acceptable & \mathrm{otherwise} \end{array}\right.$$

The practitioner’s speed is compared with the standard data used for training. Based on the SoE, the system motivates the practitioner to speed up or slow down the pose.

Angle of Deviation (AoD): Practitioners are not suggested to deviate the angle more by than 5 degrees; however, up to 8 degrees is considered to be acceptable. If the AoD is in the range of 8–16, it is considered moderate, in the range of 16–22 it is considered high, and greater than that is considered as wrong pose. Based on the AoD, the level of the practitioner is computed.
$$AoD=\left\{\begin{array}{cc} High, & \mathrm{if}\;\mathrm{if}\;\theta >16\;\&\;\theta <24\\ Moderate, & \mathrm{if}\;\theta >8\;\&\;\theta <16\\ Acceptable & \mathrm{if}\;\theta >8\\ Wrong & \mathrm{otherwise} \end{array}\right.$$

SC is computed based on AoD and SoE. The other indicators such as recall, precision, specificity, F1-score, and accuracy [129,130,131,132,133,134,135,136,137] are computed in Equations (1)–(5) below:(1)$$Recall=\frac{TP}{TP+FN}$$
(2)$$Precision=\frac{TP}{TP+FP}$$
(3)$$Specificity=\frac{TN}{TN+FP}$$
(4)$$F1-score=\frac{2\times Precision\times Recall}{Precision+Recall}$$
(5)$$Accuracy=\frac{TP+TN}{TP+FP+FN+TN}$$
where TP, TN, FN, and FP represent true positive, true negative, false negative, and false positive.

## 7. Discussion

Yoga is a globally popular form of exercise that augments physical and mental health through breathing techniques and the stretching of every body organ. It also involves the conscious process, which helps to concentrate, focus, balance, and calm down. The yoga postures are categorized into standing, sitting, lying, and prone, followed by exhalation and inhalation. The three major aspects are *pranayama* known as breathing technique, *asanas*, known as physical posture, and *dhanaya*, known as meditation. Integrating these three aspects helps to achieve multiple personality developments, including mental, emotional, intellectual, physical, and spiritual. Different types of yoga are classified based on these three aspects. For example, ashtanga yoga, a rigorous yoga pose, begins with *dhanaya* followed by *asanas*, whereas Hatha yoga, a gentle yoga pose, begins with *asanas* followed by *pranayama*.

The monitoring, detection, and analysis of yoga are essential for detecting and guiding yoga practitioners to correct their yoga postures automatically so that practitioners can perform yoga independently without the essence of yoga experts. The guidance and monitoring of yoga are crucial to ripe the full benefits of it, as wrong postures possess multiple antagonistic effects, including physical hazards and stroke. There exist very few approaches using the sensor-based yoga pose detection approach. Considering the privacy issues, the camera is not used in this approach, and this approach is computationally expensive and requires a specific location to perform yoga. Research on the smart yoga mat is still in its infancy, which requires more research and experiments to be capable of detecting, guiding, and monitoring yoga poses. Yoga practitioners usually perform yoga in an open environment, and sensor-based methods need to concentrate more on being nature-friendly rather than focusing on a specific room, such as the yoga center. Vision-based yoga pose detection methods are the common choice of many researchers due to the simplicity and availability of the data. Yoga consists of numerous postures requiring the bending of multiple parts of the body. The correct posture is defined based on the angle formed by the joint of the body parts, i.e., the bend accomplished when holding that posture. Estimating the angle between the joints of the body and finding anomalies is the most challenging task. Anomalies are usually found by comparing the participant’s posture with the expert’s posture using learning approaches. Table 4 presents some data to compare the sensor-based and vision-based approaches.

Additionally, a significant amount of research highlights that, by using yoga intervention on specific problems, such as stress, positive performance results are achieved but the reasons behind it fail to be explored. Some recent research shows poor results regarding the drawbacks and limitations of yoga interventions in addressing some specific problems, which also require appropriate reasons behind it. Similarly, the amount of data and the time frame of yoga interventions also hampers the results, which can be a major concern and needs more research to explore the reasons. These all show that research on the yoga domain is still in its infancy, suggesting that more research is essential. To date, all the research is focused only on detecting yoga poses. No approaches to detect the types of yoga exist. However, detecting the types of yoga is crucial, especially for patients and weak practitioners. For example, Hatha yoga may be more beneficial for heart patients rather than Ashtanga yoga. Every yoga type is designed to focus on numerous benefits for health. Hence, the detection of the types of yoga is as crucial as detecting yoga poses.

We designed network visualization using *VOSviewer* based on previously published articles mentioned in Figure 4. Previously, we used network visualization in the field of sleep disorders [138,139], block-chain technology [140], premenstrual psychosomatic behavioral symptoms [141], machine learning [142], deep learning [143], and smartphone addiction [144] to find the closest terms based on previous data. This method is rather useful for obtaining the precise keywords connected to any of the locations. It is a brand-new tool for studying a significant amount of previously released data. Furthermore, manually locating the nearest phrases for any sizable databases is very difficult. This program will provide a fresh software-based method for visualizing the dataset. Using clusters, it separates published articles into categories depending on words. We also designed the word cloud based on the current study (Figure 4). This showed the common words used many times in this paper.

## 8. Challenges and Future Directions of Yoga

Yoga is essential for everyone in their daily lives and is considered a natural healer. Furthermore, it has the potential to provide adequate, good, and encouraging effects in treating multiple diseases. Hence, multiple research efforts focus on treating and preventing numerous diseases, including cancer and cardiovascular diseases. However, further research is essential for long-term impacts related to biological mechanisms. Integrating IIoT technology into the practice of yoga poses several research challenges. The significant challenge lies in the data collection for the training purpose, which requires experts in the domain. Similarly, research is mostly based on yoga interventions, which lack standard guidelines for a different environment. The environmental factor is always ignored, creating a significant gap in preventing multiple diseases, including stress management. Furthermore, by creating and performing research projects with a sound methodological structure on larger sample groups, it may be required to widen the subject even further and obtain more substantial scientific proof. In summary, some of the significant challenges of yoga are:To identify the appropriate sensors and devices that can accurately capture data related to yoga practice, such as body movements, heart rate, and breathing patterns, is also challenging. These sensors should be non-intrusive and comfortable to wear during yoga practice.To develop algorithms and machine learning models to analyze the data collected from these sensors and provide valuable insights related to yoga practice, such as the effectiveness of different postures or the impact of specific breathing techniques.To ensure the security and privacy of the data collected by IIoT devices during yoga practice. Appropriate measures such as encryption and anonymization should be implemented to prevent unauthorized access and ensure data privacy.The essence of standardization in developing and implementing IIoT systems for yoga practice is to ensure the interoperability and compatibility across different devices and platforms.To ensure that the use of IIoT in yoga practice aligns with the values and principles of traditional yoga and does not compromise the integrity and authenticity of the practice.

## 9. Conclusions

This paper discussed recent advances in yoga, including the use of IIoT to identify and monitor yoga postures. To the best of our knowledge, this article reviewed yoga along with detection and monitoring for the first time and explained the significance of yoga in human health applications. This paper also discussed the different types of yoga, the importance of detecting them, and the various safety precautions to take during yoga practice. Finally, the scope of yoga’s applications and the use of the IIoT are covered. This is followed by a discussion of various challenges and future directions, and this paper is concluded. Despite numerous attempts, the integration of yoga with IIoT is still in its infancy. As a result, we present some unresolved problems that deserve in-depth consideration and future directions. As IoT technology advances, yoga practices can become more common and friendly by utilizing IoT devices.

## Figures and Tables

**Figure 1 bioengineering-10-00459-f001:**
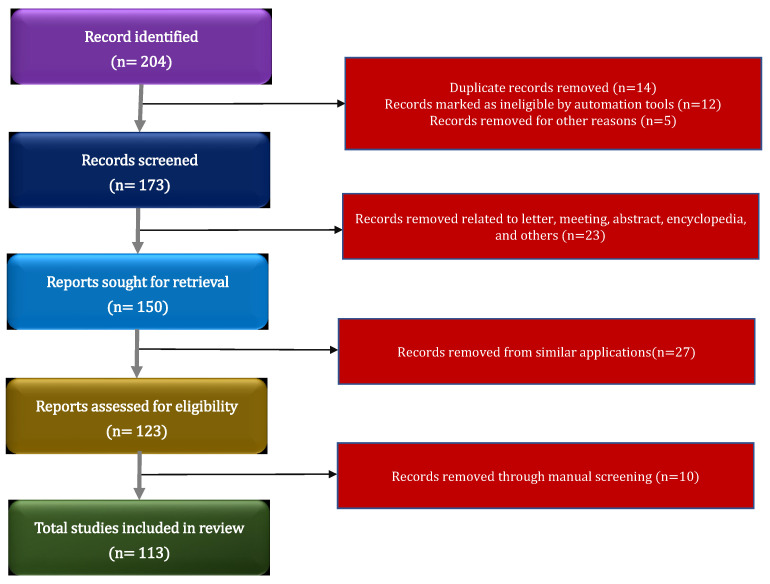
Flowchart of the current study based on PRISMA guidelines.

**Figure 2 bioengineering-10-00459-f002:**
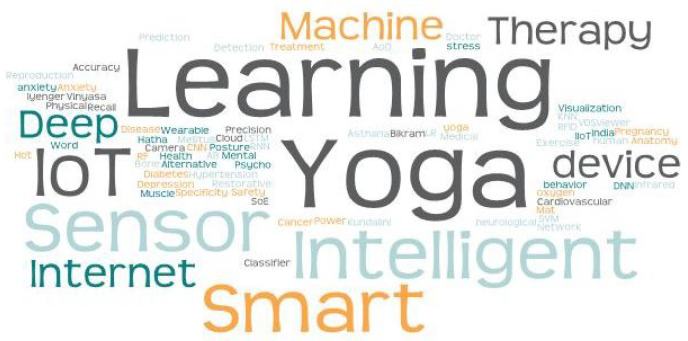
Word cloud of the current study.

**Figure 3 bioengineering-10-00459-f003:**
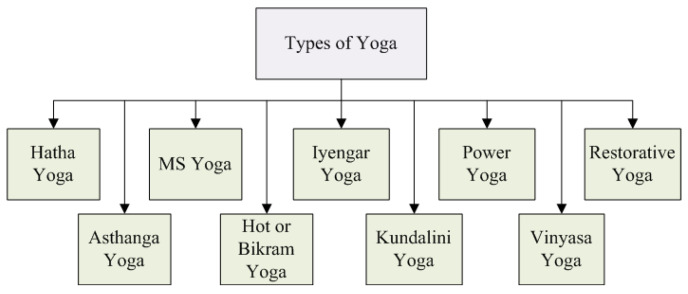
Some of the most popular types of yoga.

**Figure 4 bioengineering-10-00459-f004:**
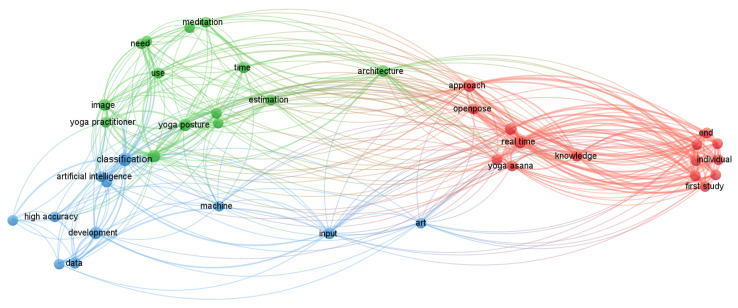
Network visualization of yoga with intelligence. This shows the closest terms used in the previously published articles related to yoga with intelligent techniques. These terms are useful for collecting data for upcoming research in this area.

**Table 1 bioengineering-10-00459-t001:** Comparison of the existing literature and our survey. *√* indicates topics covered in the literature.

		[19]	[20]	[21]	[22]	[30]	[23]	[24]	[25]	[27]	[29]	This Survey
Yoga Background	General Background	*√*	*√*	*√*	*√*	*√*	*√*	*√*	*√*	*√*	*√*	*√*
	Types of Yoga											*√*
	Safety Measures											*√*
	Yoga vs. Pranayama	*√*										*√*
	Challenges	*√*		*√*	*√*							*√*
Detection of Yoga	Sensor-based					*√*						*√*
	Vision-based					*√*						*√*
	Hybrid-based											*√*
Application	Single		*√*				*√*	*√*	*√*	*√*	*√*	
	Multiple	*√*		*√*	*√*							*√*
New Perspectives												*√*

**Table 2 bioengineering-10-00459-t002:** Comparison of different yoga types.

Reference	Type of Yoga	Descriptions
[32]	Hatha Yoga	It employs physical techniques to retain and channel vital force or energy. It aligns, cleanses, and calms your mind, body, and spirit, allowing you to achieve deeper levels of meditation and spiritual awareness. It helps to increase stamina, flexibility, range of motion, and balance; as well as to decrease tension and promote mental calm.
[33]	Asthanga Yoga	Physically demanding and challenging style. It is a strenuous practice that promises to increase physical stamina and flexibility.
[34,35,36]	Hot or Bikram Yoga	It uses a fixed 26-posture series and two breathing exercises to help simulate the temperature. Bikram yoga has positive benefits on metabolic markers such as blood lipids, insulin resistance, and glucose tolerance.
[37,38]	Iyengar Yoga	It is slower than Ashtanga or Bikram, but this form of yoga still calls for a great amount of attention to attain correct posture alignment and the capacity to hold the asanas for extended periods of time. Persistent lower back pain significantly decreased in trial participants who practiced Iyengar yoga.
[39]	Kundalini Yoga	In the form of chanting or song, it combines movement, breath, and sound. Kundalini is meant to awaken the shakti, the spiritual force that resides at the base of your spine. It can positively impact our mood, concentration, blood pressure, metabolism, and strength, among other health benefits.
[31]	Power Yoga	Less structured and more open to personal interpretation by teachers. Generally, more physical and performed at a faster tempo than other kinds of yoga. Increases flexibility while also strengthening the muscles.
[40]	Restorative Yoga	Help with stress relief because lying in these postures for extended periods of time allows you to listen to your body’s signals and focus your mind. Nurses working night shifts reported that group restorative yoga sessions significantly reduced their psychological and physical stress reactions.
[33]	Vinyasa Yoga	Also known as flow yoga or vinyasa flow. The word “vinyasa”, which means “special spot”, generally means connecting breath and movement. Vinyasa or flow is frequently used with slow, dynamic, or attentive movement to denote the intensity of practice and is more appropriate for newcomers to yoga than for those who have been practicing it for years.

**Table 3 bioengineering-10-00459-t003:** Detection of yoga using various IIoT approaches and their comparison.

References	Approaches	Descriptions
[13,14,15,85,86,88,89,90,91,92,93,94,95]	Sensor-based approach	Multiple sensors are used to detect yoga postures, including wearable, infrared sensors, RFID, and smart mat.
[96,97]	Vision-based approach	Relies on the camera for the input, which is further processed using intelligent approaches for the detection of the yoga postures.
[98]	Logistics regression	An extension of ordinary regression; it is a powerful and popular technique for supervised classification for modeling a dichotomous variable for an associated label.
[99]	Adaboost	An ensemble method to combine weak classifiers to create a powerful classifier. To attain high accuracy for the model, it continues to add learners until a robust classifier is reached.
[100,101,102]	Random forest	In RF, each tree is reliant on values from a random vector that was randomly sampled and had a uniform distribution across all of the forest trees.
[103]	Support vector machine (SVM)	It has two classifiers and is an SVM classifier. Nonetheless, a multiclass SVM is widely used because most issues involve multiple classes.
[3,104]	K-nearest neighbor (KNN)	KNN saves all potential examples and categorizes them according to their similarities. It is primarily used with the pattern recognition method.
[105]	Deep learning-based methods	Deep learning is essentially based on ANN and it can be compared to the human brain.
[106,107,108,109]	AutoEncoder	A rich and versatile framework for discovering the salient features of data in an unsupervised manner. Used to drive the learning of a deep illustration of the volumetric human body structure.
[103,110,111]	Convolutional neural networks (CNN)s	A great choice because they have proven to have a significant amount of potential for pose classification tasks. They can be trained directly on pictures or on key human skeleton joint locations.
[112]	Recurrent neural networks (RNNs)	RNNs are useful for processing sequential data since they preserve a neuron’s prior data. RNNs have difficulty remembering the initial steps necessary to forecast the current task when there are too many intermediate steps in a yoga asana.
[113]	Long short-term memory (LSTM)	A well-known RNN called an LSTM has the ability to naturally remember knowledge or data for sufficient lengths of time. The LSTM algorithm employs three gates: input, update, and forget. Resultantly, an LSTM will selectively ignore or recall the learned information.
[114,115,116,117,118]	Deep neural networks (DNNs)	DNNs have demonstrated exceptional performance on visual classification functions. DNNs can capture the complete context of every body joint since each joint regressor uses the entire image as a signal.
[119,120,121,122,123,124,125,126,127]	Hybrid approaches	Several algorithms make use of hybrid models. For example, SVM and Inception V3 are hybrid algorithms. Another study classified data using a hybrid 798 CNN–LSTM layer after extracting key points using OpenPose.

**Table 4 bioengineering-10-00459-t004:** Comparison of the performance of the sensor-based and vision-based approaches.

Ref.	Asana (No.)	Method	Performance	Method
[86]	6	CNN, LSTM	99.91%	Sensor-based
[89]	6	CNN, LSTM	89.29%, 96.31%	
[90]	5	CNN	96%	
[88]	10	XBoost	99.2%	
[124]	10	RF, KNN, SVM, DT	94.28%	Vision-based
[14]	26	DCNN	99.99%	
[110]	42	CNN	98.93%	
[99]	5	Adaboost	94.78%	
[101]	10	RF	96.47%	
[103]	4	SVM	94.9%	
[3]	4	KNN	93.1%	
[114]	17	LSTM	97.7%	
[105]	8	CNN, SAE	90%, 70%	

## Data Availability

Not Applicable.

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
