# Peer review of "Yoga Meets Intelligent Internet of Things: Recent Challenges and Future Directions"

_bioengineering, 2023, doi:10.3390/bioengineering10040459_

Round 1
Reviewer 1 Report
This paper carried out a provide a survey on integrating yoga with IIoT. The paper also discusses the multiple types of yoga and the procedure for the detection of yoga using IIoT. Additionally, the paper highlights various applications of yoga, safety measures, various challenges, and future directions. This paper must be re-outlined in order to clarify the main ideas and the contribution. Some sections must be added or improved:
1) I believe the authors must establish a set of research questions to be answered in this survey. The academic contribution could be more clear. The three research questions, presented in section 1.1. Methodology, are very general. I suggest describing the systematic process of this survey in a PRISMA diagram.
2) In the Related Work section is very important to realize a comparative analysis of this proposal with other reviews in order to clarify the differences.
3) Authors establish three main contributions, but I believe the second and third points are findings of the review. The authors combine a review of apps, a review of approaches for monitoring and detection of yoga, and types of yoga. What are the relationships among them? A systematic literature review is very specific.
4) Authors mention 131 papers were selected, I think the inclusion and exclusion criteria must be explicitly described.
5) I think Section 2 is not very relevant to this survey.
6) Some most interesting aspects of Yoga is to know the following:
a. What kind of IIoT devices are the most used? Why?
b. What kind of Machine/Deep learning algorithms are the most frequent in the literature? What values are presented for accuracy, precisión, recall, F1-measure, ROC, to mention but a few.
7) The Discussion section is very short. In this section, the main findings of the review must be presented and should be emphasized.
8) The validity of this literature review might be affected by two threats: inadequate selection of primary studies and incorrect data extraction. What authors can argue about these aspects?
9) The Conclusions section is not good and the authors have not properly commented on all outcomes of this work.
In general terms, the paper requires improving its structure. Some major changes must be done to complete the description of the work and the research. The research topic is relevant, however, the authors must improve the structure and presentation of this paper. So, my recommendation is “Accept with major revision”. It is not clear what is the innovation and the “academic” value of this proposal in comparison with other review initiatives.
I suggest following the guidelines proposed by Brereton [1] for elaborating systematic literature reviews.
References
[1] Pearl Brereton, Barbara A. Kitchenham, David Budgen, Mark Turner, Mohamed Khalil. (2007). Lessons from applying the systematic literature review process within the software engineering domain, Journal of Systems and Software, Volume 80, Issue 4, Pages 571-583, ISSN 0164-1212, https://doi.org/10.1016/j.jss.2006.07.009
Reviewer 2 Report
(1)Please check the manuscript carefully to remove the typos, improve the language and format.
E.g.
-The data of the 1st affiliation are not complete.
-The 4th affiliation should be listed in another line.
-Some characters are thin or wide, e.g. Fig. 2. Please adjust the aspect ratio of the characters.
-[? ]. (Line 539, Page 12)
...
(2)Some paragraphs are too long and difficult to follow, e.g. Para 2 Section 2, Section 3.1. Please divide them into several short paragraphs to improve the readability.
(3)The authors should use more table to compare the types, such as the different types in Section 2,
(4)The length of this paper is too long, although it is a survey. Some well-known knowledge, e.g. Section 5, can be removed or shortened/condensed, since they can be easily found in textbooks, and are not firstly proposed in this paper.
(5)The authors should use more figures, tables and qualitative equations for the statements in Section 5.
Reviewer 3 Report
The authors presented a comprehensive survey on integrating yoga with IIoT. The authors highlighted various applications of yoga, safety measures, various challenges, and future directions. The authors also discussed the state-of-the-art research in addressing the challenges and future direction regarding yoga and its integration with IIoT. The discussion mainly binds to overviewing, classifying, and analyzing yoga and its integration with IIoT for monitoring and automatic guidance without the essence of the yoga expert. The authors need to address the following comments.
A. Related work section needs further comprehensive discussion.
B. The authors discussed the detection of Yoga using Intelligent Internet of Things. The authors should discuss the outcomes, challenges more comprehensively. A summarized table will be beneficial to understand the tools and technology and challenges.
C. The authors also discussed the Machine Learning based Approaches. Need to summarize the ML algorithms to show the main features and obtained results.
D. The authors should concisely state the research challenges.
E. What is the outcome of research. It is not clear.
F. The discussion is very general. We need further comprehensive discussion and scientific soundness.
G. The paper lacks novelty. The authors need to justify.
Reviewer 4 Report
- Unfortunately, the novelty of the approach is not well motivated. As such, the introduction could be improved to highlight what the authors consider as the main novelty of their work specifying the main difference with the works in literature.
- An important aspect is handling data heterogeneity typical in the faced application scenario. Data integration is a key issue in this context. Distributed data sources can be heterogeneous in their formats, schemas, quality, access mechanisms, ownership, access policies, and capabilities. Data integration is the flexible and managed federation, analysis, and processing of data from different distributed sources. Data integration is becoming as important as data mining for exploiting the value of large and distributed data sets that today are available. Distributed processing infrastructures such as Cloud, Grids and peer-to-peer networks can be used for data integration on geographically distributed sites. See the following reference:
https://link.springer.com/chapter/10.1007/978-3-540-30470-8_27
- In the faced context is particularly relevant also considering the issues arisen by the architectural choices. Collecting data from users’ devicesis key in mart health applications that are executed over mobile devices, like for example, sensors distributed over patients body. In a mobile network, is key ensuring efficient routing, resource allocation, and energy management. Accordingly, authors should highlight in the Introduction or Related Work Section the importance of addressing mobile ad-hoc solutions tailored to data intensive smart health applications, as reported for example in the following reference:
o https://www.sciencedirect.com/science/article/abs/pii/S1574119217300263
Round 2
Reviewer 1 Report
The authors have adressed all the comments. The content of this paper has been improved
Author Response
Thank you for your appreciation and Support.
Reviewer 2 Report
The length of this paper is too long. Some well-known knowledge and unnecessary experiments can be removed or shortened/condensed, since they can be easily found in textbooks, and are not firstly proposed in this paper.
Author Response
Thank you for your kind advice. We have condensed some sections to shorten the length of the paper and reduced it from 31 pages to 26 pages. As you suggested, we removed some sections from section 5 and condensed other sections too.
Reviewer 3 Report
Thanks for revising the manuscript and addressing the comments.
Author Response
Thank you for your appreciation and support.
Reviewer 4 Report
The authors addressed all the issues and comments of my previous review. Therefore, the paper can be accepted as is.
Author Response

(The authors gave the same response as above.)
